# CAN LARGE LANGUAGE MODELS HELP WITH MODEL COUNTING?

## ABSTRACT

Model counting is a fundamental problem in computer science with several applications, ranging from mutation modeling in DNA to statistical physics. Instead of finding one solution for a task at hand, in model counting we want to compute the exact number of different solutions for this task. While large language models (LLMs) have impressive performance in different reasoning tasks, their effectiveness has been focused in the context of optimization and decision problems. In this paper, we bridge this gap studying the capabilities of LLMs in model counting for combinatorial problems. Using the popular Sudoku puzzle as an illustrative example, we evaluate how good LLMs are in counting the number of solutions for different Sudoku puzzles. We show that, despite having decent performance at first, LLMs are fragile to modifications in the problem encoding. On top of that, we also study how different representations of the problem impact performance. In particular, we preprocess our formulas and transform them into d-DNNF formulas. This is an important fragment of propositional logic for which the model counting problem is trivial. With this simple fragment, the performance of the reasoning LLMs improves, indicating that they might be capable of counting simpler problems with chain-of-thoughts, although not consistently. Last, we also study whether LLMs can generate Python programs that compute the exact model counts. Unsurprisingly, while LLMs struggle to count by themselves, they are much more reliable when creating code to do this job.

## 1 INTRODUCTION

Many computational problems can be expressed as reasoning tasks over logical rules (Harrison, 2007; 2009; Cook, 2021; Marques-Silva, 2025; Avigad, 2024; Hanna et al., 2024). The area of *automated reasoning* aims at solving problems representing in such way. Automated reasoning usually focuses on *decision problems*: for example, deciding whether a Boolean formula is satisfiable (SAT). Despite its computational hardness, SAT has been a cornerstone of symbolic AI (e.g., Kautz & Selman, 1992) and theoretical computer science (Knuth, 2015; 2023), with an imminent role in complexity theory (Cook, 1971; Levin, 1973).

With the recent progress in large language models (LLMs), the reasoning capabilities of LLMs has gathered a lot of attention. LLMs can already help with many logical reasoning tasks with surprising effectiveness (e.g., Li et al., 2022; Romera-Paredes et al., 2024; DeepSeek-AI et al., 2024). Even in symbolic AI, LLMs have shown potential to help existing solvers or to expand the use cases for these tools (e.g., Katz et al., 2024; Corrêa et al., 2025).

However, LLMs have not yet been studied in the context of *quantitative reasoning* — where the goal is not to find a solution but to *count the number of solutions*. Propositional *model counting*, also known as *#SAT*, generalizes SAT by asking how many assignments satisfy a given Boolean formula. This problem is significantly harder than SAT itself: Toda's theorem shows that solving #SAT efficiently would collapse the polynomial hierarchy (Valiant, 1979; Roth, 1996; Bacchus et al.,

2003; Toda, 1991). As the number of solutions can be exponential, brute-force enumeration is infeasible. Despite this complexity, model counting is central to a wide range of domains: from DNA analysis and statistical physics (Grohe & Thurley, 2011; Nagy et al., 2024), to cryptanalysis (Beck et al., 2020; Girol et al., 2024), quantum circuit verification (Mei et al., 2024a;b), and explaining neural networks (Shi et al., 2020). In many of these applications, counting provides quantitative information that decision procedures alone cannot capture.

LLMs show promising abilities in automated reasoning tasks, often by combining natural language understanding with tool use or chain-of-thought prompting (Yao et al., 2023; Szeider, 2025). However, these successes have so far been limited to decision or optimization problems. Recent theoretical results even suggest that LLMs, and transformers more broadly, have very limited arithmetic capabilities (Peng et al., 2024). This raises a natural question: can LLMs provide any practical advantage for model counting tasks such as #SAT?

In this paper, we address this topic from a practical perspective and study the capabilities of LLMs in model counting for combinatorial problems. To the best of our knowledge, this has not been attempted before and there are no studies into that direction to date. To analyze this, we study the performance of LLMs in a well-known combinatorial puzzle: Sudoku. Sudoku is an appealing testbed because it has a simple formulation, a large body of prior work on propositional encodings (Lynce & Ouaknine, 2006; Bright et al., 2020), and a clear notion of solution counts. By expressing Sudoku instances as Boolean formulas, we can evaluate whether LLMs are able to estimate or compute the number of satisfying assignments.

Our focus, however, is not purely whether the LLMs can estimate the number of solutions to a Sudoku puzzle. Instead, we study how different inputs and prompting techniques change the performance of the models. In other words, we study a gradient of strategies for model counting. First, we evaluate the *end-to-end model counting* performance of the models (i.e., simply prompting the model to return the answer). Then, we analyze the impact of the model having access to the information of the combinatorial planning it is solving—i.e., when it is told that the formula at hand corresponds to a Sudoku puzzle and when it is not. We also investigate what happens when the problems are obfuscated, a common strategy in the evaluation of LLMs on symbolic tasks (Valmeekam et al., 2023b). But more interestingly, we study the impact of *tractable fragments* for counting. It is well-known that for certain families of logic formulas (based on their structure) we can count the number of satisfying assignments in polynomial time, and that CNFs can be converted into these formats (Darwiche & Marquis, 2002). In our experiments, we convert the original Sudoku formulas (in CNF) to these formats, and investigate whether this improves the performance of the LLMs. Last, but not least, we experiment with *code generation*: instead of requesting the LLM to simply output an answer, we prompt it so it generated a Python program that counts the number of solutions.

Perhaps unsurprisingly, the code generation method outperforms all other methods. Moreover, our results suggest that the LLM is aware of how to exploit the tractable fragments, even though we never inform them that the input formulas fall into a tractable case. Overall, our paper gives a first broad study on the capabilities of LLM for model counting.

**Summary of Results.** Our main findings are the following:

1. We show that, despite having decent performance at first, LLMs are very fragile to modifications in the problem input (e.g., formula permutation or larger instances). This indicates that the LLMs might have seen the original problems (which have been used in competitions) during training.

2. We study how different representations of the Sudoku problem impact performance. In particular, we preprocess our formulas and transform them into formulas on which counting is efficient (d-DNNF). With this simple fragment, the performance of the reasoning LLMs improves considerably, indicating that they might be capable of counting simpler problems with chain-of-thoughts, although not consistently.

3. Finally, we analyze whether LLMs are able to generate efficient Python code to solve these problems. The generated code is surprisingly accurate, although not perfect. Still, the results are somewhat impressive, given that the functions produced by the larger models (e.g., DeepSeek R1) seems conceptually different from that in the literature or existing solvers.

## 2 BACKGROUND AND PROBLEM DESCRIPTION

In this section, we briefly describe the underlying problem of propositional logic, counting therein, and the Sudoku puzzle in propositional logic. For a comprehensive introduction, we refer to introductory literature (Biere et al., 2009; Kleine Büning & Lettman, 1999).

### 2.1 PROPOSITIONAL SATISFIABILITY

We assume sets to be finite and we let $2^X$ be the *power set of $X$* for a set $X$. Let $U$ be a universe of propositional variables. Classically, when defining propositional formulas in logic textbooks, we allow arbitrary formulas without any particular normal form. In practice, solvers employ the so-called conjunctive normal form (CNF), where one can check extremely efficiently whether the formula is already satisfied (Moskewicz et al., 2001). Restricting the input to CNF formulas is a fairly weak restriction. Using the Tseitin transformation (Tseytin, 1983), we can efficiently transform any propositional formula into CNF without affecting its model count at the expense of introducing functionally-depended auxiliary variables. Therefore, we provide only definitions for CNF. A *literal* is a variable $x$, from the universe of propositional variables, or its negation $\neg x$. We call $x$ *positive* literal and $\neg x$ *negative* literal. A *clause* is a finite set of literals, interpreted as the disjunction of these literals. A propositional *formula $F$* in *conjunctive normal form (CNF)* is a finite set of clauses, interpreted as the conjunction of its clauses. We let $\mathrm{var}(F)$ and $\mathrm{lits}(F)$ be the set of the variables and set of literals, respectively, that occur in $F$. An *assignment* is a mapping $\tau : X \rightarrow \{0, 1\}$ defined for a set $X \subseteq U$ of variables. The formula $F$ *under assignment $\tau$* is the formula $F_\tau$ obtained from $F$ by (i) removing all clauses $c$ that contain a literal set to 1 by $\tau$ and then (ii) removing from the remaining clauses all literals set to 0 by $\tau$. An assignment $\tau$ *satisfies* a given formula $F$ if $F_\tau = \emptyset$. For a satisfying assignment $\tau$, we call the set $M$ of variables that are assigned to true by $\tau$ a *model* of $F$ and let $\mathrm{Mod}(F)$ be the *set of all models* of a formula $F$. Furthermore, we define the *model count* $\mathrm{mc}(F)$ as the number of models of the formula $F$.

**Example 1.** *Consider the following formula $F = (a \vee \neg b \vee d) \wedge (\neg a \vee \neg c \vee \neg d) \wedge (\neg a \vee \neg d) \wedge (b \vee c) \wedge (d \vee e) \wedge (\neg b \vee \neg e)$. Clearly, the set $\{c, d, e\}$ is a model of $F$ and in total $F$ has 6 models.*

Besides CNFs, we need another normal form, namely, *d-DNNF* (Darwiche & Marquis, 2002), which have the following restrictions. A propositional formula (not necessarily in CNF) is in negation normal form (NNF) if the negation occurs only in front of variables, and otherwise only conjunction and disjunction occur as connectives. A NNF is in *decomposable (DNNF)* if conjunctions do not share any propositional variables, and *deterministic (d-NNF)* if disjunction do not share models (jointly inconsistent), and *smooth (s-NNF)* if, for each conjunction, its children (in the parse-tree) are composed of the same set of variables.

Commonly, the input for SAT instances are given in the so-called DIMACS-CNF format (Trick et al., 1993), which is a rudimentary text-format solely based on representing the variables by indices. There variables are consecutively numbered by positive integers. A problem line indicates the number of occurring variables and clauses, respectively. The remaining lines indicate clauses consisting of decimal integers separated by space and terminated by 0. The tooling landscape for propositional satisfiability is ample, with numerous SAT solvers that compete in competitions every year (Codel et al., 2025) among them particularly successful ones (Biere, 2019; Biere et al., 2020). A Python library to access state-of-art SAT solvers is well-established and widely used for problem solving (Ignatiev et al., 2018). Very recently, a system enhance LLMs by symbolic solvers using

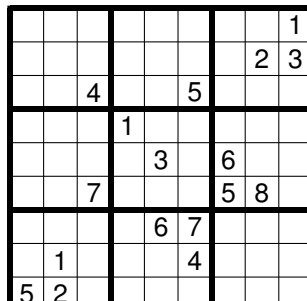 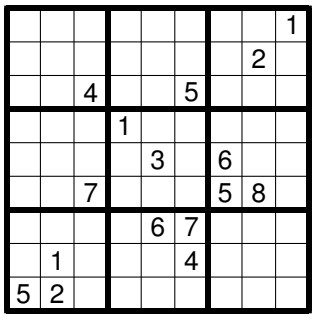 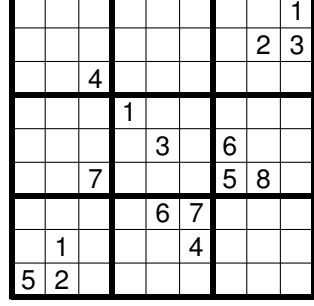

Figure 1: Sudoku with 1 solution (left), 38911 solutions (middle), and 83525 solutions (right).

the Model Context Protocol (MCP) has been suggested for SAT as well (Szeider, 2025) demonstrating that LLMs can be used to capture natural language understanding and turn this into encodings, which can then be solved by state-of-the-art solvers.

From a computational perspective, we know that model counting is one of the hardest problems.

**Proposition 1** (Valiant, 1979). *#SAT is #P-complete.*

Recall that SAT (its decision version) is "only" NP-complete (Cook, 1971). With model counting instead we solve any problem located on the Polynomial Hierarchy (PH), meaning an arbitrary number of oracle calls interleaved, following from an immediate consequence of Toda's Theorem (Toda, 1991). #SAT remains computationally very hard on several classes of propositional formulas (Valiant, 1979; Dahllöf et al., 2005), but becomes easy on numerous others (Darwiche, 2001; Darwiche & Marquis, 2002).

**Proposition 2** (Darwiche, 1999). *While d-DNNFs can still have exponentially many solutions, #d-DNNF is polynomial-time solvable.*

## 2.2 SUDOKU PUZZLE AND ENCODING

*Sudoku* is a logic-based, combinatorial number-placement puzzle (Felgenhauer & Jarvis, 2006; Delahaye, 2006; Ercsey-Ravasz & Toroczkai, 2012). Classically, we have a 9x9-grid consisting of nine 3x3-sub-grids, called regions. Each entry (also called a cell) in the grid may contain a digit from 1 to 9. Initially, the grid is partially filled, meaning some entries are preassigned and others are left blank. The task is to fill-in digits such that each column, each row, and each region contains all the digits from 1 to 9 exactly once. Popular Sudoku puzzles usually contain a single solution, but of combinatorial interest also grids with multiple solutions exist. The Sudoku problem has received attention in Constraint Programming (Simonis, 2005) and SAT (Lynce & Ouaknine, 2006) to the extent of solving Sudoku versions on extremely large grids (255x255-grids with numbers instead of digits). A comprehensive introduction to the topic, including is source code to convert Sudokus into CNF-formulas, is available online (Junttila, 2020).

## 3 RELATED WORK

Research on the reasoning abilities of large language models (LLMs) has gathered a lot of attention in recent years. We briefly survey some previous relevant results below.

**LLMs and Arithmetic.** Previous work highlights the difficulties LLMs face with basic counting or arithmetic tasks. For example, Fu et al. (2024) and Centulani (2024) show that even simple tasks such as counting characters or performing small calculations are challenging to LLMs. These practical results are supported by recent theoretical work: Peng et al. (2024) argue that transformers

have very limited arithmetic capabilities in principle. (Note that this is a different notion of counting from our paper: in our case, we are interested in *counting solutions*, and not counting characters or arithmetic tasks.)

**Symbolic Reasoning with LLMs.** Despite these limitations, LLMs have been increasingly used in symbolic AI tasks. They have been employed to assist or extend classical solvers, either by generating heuristics (Corrêa et al., 2025) or by guiding search processes in planning (Katz et al., 2024). Similarly, Szeider (2025) proposes coupling LLMs with SAT solvers via the Model Context Protocol (MCP), allowing natural language input to be translated into formal encodings. In similar directions, LLMs have shown promising results in program synthesis (Li et al., 2022), in mathematical discovery (Romera-Paredes et al., 2024), and even in broader multimodal reasoning tasks (DeepSeek-AI et al., 2024).

**Combinatorial Reasoning.** Model counting has been a topic of interest for a long time (Valiant, 1979; Toda, 1991; Roth, 1996). In the context of LLMs, however, there is little prior work. Our work is, to the best of our knowledge, the first to evaluate LLMs on exact model counting, bridging the gap between classical symbolic AI and emerging LLM-based reasoning.

## 4 EXPERIMENT AND LIMITATIONS

To evaluate the model counting capabilities of current LLMs, we tested different models in a Sudoku benchmark. Our experiments test different aspects of model counting: end-to-end model counting, where we simply ask the models for a solution; end-to-end model counting on permuted tasks, where we reduce the chance that the model saw the solutions during training; end-to-end model counting on simpler tasks, where it is known that polynomial-time algorithms exist to compute the exact counting; and code generation, where we expect the model to be able to produce (Python) code to solve the tasks.

Our code, experimental raw data, and benchmarks will be made publicly available in case of acceptance.

**Setup.** In our experiments, we have tried two different model families: Gemini (Google et al., 2023) and DeepSeek (DeepSeek-AI et al., 2025). From the Gemini family, we tested Gemini 2.5 Flash and Gemini 2.5 Pro; from DeepSeek, we tested DeepSeek R1. Both Gemini 2.5 Pro and DeepSeek R1 are thinking models, so we expect them to perform better. We decided to include Gemini 2.5 Flash to see how a non-thinking model performs.

All our experiments were run via the official APIs of each model. For the Gemini models, our experiments used the free tier of their API so there were no monetary costs. The experiments with DeepSeek R1 cost on average $0.03 USD per instance. We use temperature 0.6 for all our runs.

**Metrics.** Our evaluation focuses on *accuracy*, meaning the number of problems that the models answered exactly right. For wrong solutions, we do not take into account how far from the correct solution the returned value was. We also analyze the number of reasoning tokens produced by the different methods tested. This can help us to estimate the computational effort required by each one of our approaches.

**Datasets.** We use the Sudoku dataset from the model counting competition 2022 (Fichte & Hecher, 2022; 2025), which is a commonly used domain and serves as a testbed example. Moreover, Sudokus have also been used for practical experiments to evaluate reasoning models (Seely et al., 2025; Kumar, 2025). The dataset contains instances for 4x4 and 9x9 Sudokus. They correspond to different partially filled Sudoku grids with different number of entries already filled in each instance. The dataset also includes alternative SAT encodings for each task.

The value of the answers ranges from 1 (e.g., a 9x9 Sudoku with 17 filled entries has only one solution) to 6,670,903,752,021,072,936,960, the number of solutions in an empty 9x9 Sudoku. The basic encoding (Lynce & Ouaknine, 2006) consists of the following parts represented as CNF formula: there is at least one digit in each entry; each number appears at most once in each row; each number appears at most once in each column; each number appears at most once in each region; there is at most one number in each entry; each number appears at least once in each row; each number appears at least once in each column; each number appears at least once in each region. The dataset contains a variation of the encoding, called the full encoding. This full encoding includes also redundant constraints that increase the size of the instance, but simplify the practical reasoning. It is known (empirically) that they reduce the runtime by solvers as they add shortcuts in the reasoning (Fichte & Hecher, 2022; 2025).

**Prompts.** As we have several types of experiments, we also use different prompts (for end-to-end counting, code generation, etc.). We do not include the full prompt in the submission to avoid possible data contamination issues for the final version of the paper. In case of publication, we will include examples of the complete prompts in an appendix.

Overall, all prompts have the following general format:

- an instruction telling the model what to do (e.g., return an exact answer, generate code) and the formatting requirements,

- an explanation of the syntax used to encode the input formula, and

- the instance to be solved.

For the code generation experiments, we explicitly ask for a Python code that receives the task as standard input.

We never mention the complexity of solving the problems. In other words, for inputs in d-DNNF, we never explicitly say that they can be solved in polynomial time with simple strategies.

We also do an ablation by including the information that the model is solved a Sudoku problem, and also by removing this information so the model must figure out what problem it is. The prompt where we explicitly say that the formula corresponds to Sudoku is called our Base prompt; the one without saying the domain of the formula is called the Generic prompt.

### 4.1 END-TO-END MODEL COUNTING

We first analyze the performance of the LLMs in the original instances used in the competition. Table 1 shows the results under the columns "Original". We can see that, as expected, the thinking models outperform Gemini 2.5 Flash, the non-thinking one. Surprisingly though, the models performed better on average using the Generic prompt. This seems to indicate that the Sudoku hint makes the model get lost in identifying what each variable and each clause represent, distracting the model from actually coming up with the model count.

Overall, the results are unexpected. Some counts are very large, so the LLMs could not enumerate solutions one by one. Looking at the thoughts summaries of the thinking models (the complete reasoning tokens are not available via the API), the models seem to be approaching the counting from the right perspective, and the inference to the correct answers is sound.

However, we suspect that the solutions to the original instances have only been memorized by the model. To verify if this is the case, we permuted the original formulas by altering the labels of the variables, and the order of the clauses. This sort of obfuscation is common when evaluating other LLMs in symbolic reasoning (Valmeekam et al., 2023a). It essentially tries to minimize the possibility that the model is simply retrieving a solution seen during training.

Table 1: Number of solved instances by different models with different prompt formats and encodings. "Original" means the original Sudoku benchmark from the model-counting competition; "Permuted" are the same instances but permuted to obfuscate the original task. The "Base" prompt is the basic prompt explaining that the instance comes from a Sudoku benchmark, while the "Generic" prompt does not state which domain the instance is from.

| | Original | | Permuted | |
| --- | --- | --- | --- | --- |
| | Base | Generic | Base | Generic |
| DeepSeek R1 | 66.7% | 72.2% | 11.1% | 5.5% |
| Gemini 2.5 Pro | 55.5% | 55.5% | 22.2% | 5.5% |
| Gemini 2.5 Flash | 22.2% | 44.4% | 0.0% | 27.7% |

Table 2: Number of solved instances by different methods using d-DNNF. Showing the comparison to end-to-end model counting with CNFs using the permuted instances. Using "Base" prompts, informing the model that these are Sudoku problems.

| | End-to-End | | Python Code | |
| --- | --- | --- | --- | --- |
| | CNF | d-DNNF | CNF | d-DNNF |
| DeepSeek R1 | 20% | 40% | 60% | 80% |
| Gemini 2.5 Pro | 10% | 70% | 90% | 90% |
| Gemini 2.5 Flash | 0% | 70% | 100% | 80% |

The header "Permuted" in Table 1 shows the results. Indeed, we can see that there was a significant drop in performance using these new tasks. This indicates that, indeed, the impressive performance on the original instances was probably due to the models being trained on their solutions.

Gemini 2.5 Flash had a 0% performance in the permuted instances with the Base prompt, and 27.7% with the Generic prompt, which seems an anomaly.

### 4.2 COUNTING ON EASIER FRAGMENTS

Our next set of experiments converts all formulas to d-DNNF first, and then asks the models to count the number of solutions. We use only the Generic prompt for this experiment. As the conversion from CNF to d-DNNF might take exponential time, only half of the instances of our dataset could be converted. These correspond to the smallest and easiest half of the set. Nonetheless, this is enough to give us some insights.

Table 2 shows the results (column "End-to-End"). The values for CNF here are computed only over the permuted instances that could be also translated to d-DNNF. We can see that the models improve when the d-DNNF is given as input. In particular, the Gemini models have a large boost in performance. This supports our hypothesis that the model can identify the tractable fragment and, following that, the necessary operations are simple so the LLM can perform them. However, the models' performance is still not great, given that this is a subset of our benchmark consisting of smaller instances.

### 4.3 CODE GENERATION

Our last experiment asks the LLMs to create Python code computing the model count, instead of computing the solution itself. Our hypothesis is that the LLM can generate the domain-dependent

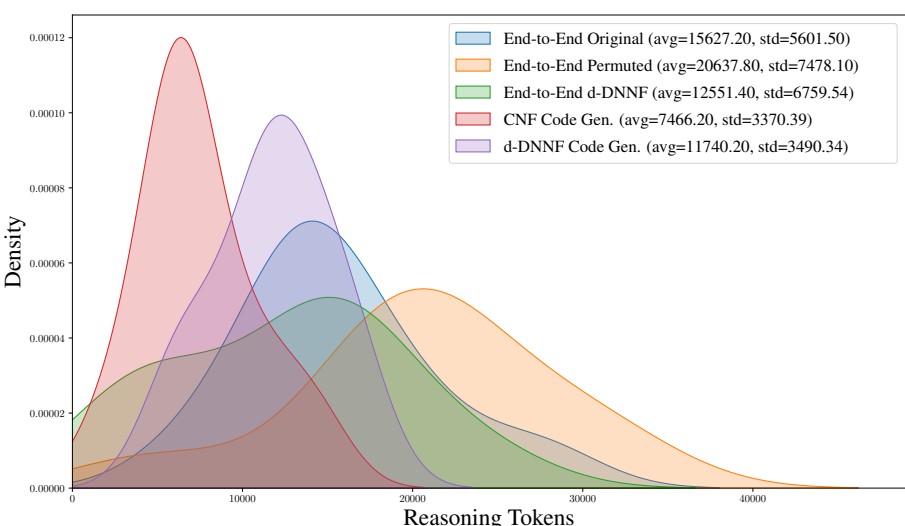

Figure 2: Distribution of total number of reasoning tokens for different methods with DeepSeek R1. Considering only those instances that were used by all experiments.

necessary code, but it fails to perform the operations itself without tools. To see if the LLM can use domain-dependent knowledge in form of code, we use the Base prompt here.

Table 2 again shows the results (column "Python Code"). It is noticeable that this approach has a significant impact on performance for both format encodings. For the CNF encoding, the gain is expressive while for d-DNNFs it is simply incremental. But this supports our claim from the previous sections: the d-DNNF encoding already allows the LLMs to compute the counts themselves, while the CNF encoding is too general and challenging to be computed used chain-of-thought. However, as the dataset used for this particular experiment is not very challenging (as it contains the instances that could be converted to d-DNNF), it is possible to for a Python program to compute the counts.

Looking more closely at the generated programs, we see some interesting patterns. For CNF instances, the programs usually rely on classical SAT solving methods, like unit propagation. They do not contain more modern ideas, like watched literals. Basically, the programs rely on enumerating the solutions and then counting how many were there. As the instances testes are small, this is a feasible technique. For the d-DNNF instances, the code often corresponds to the known polynomial algorithms. There are certain corner cases that are not properly handled. This leads to numerical problems which in turn produces the wrong answers. The generated code also does not properly handle these exceptions. Nevertheless, it is remarkable how well carried out the code is and it also works correctly on a majority of the instances.

The unsolved instances are all due to Python errors. One alternative is to compute a pool of candidates, similarly to FunSearch (Romera-Paredes et al., 2024) and other best-of-n approaches. We believe that the test with CNF instances had better performance because the functions implemented (unit propagation, backtracking) occurred more frequently in the training data.

We also tried the code generation methods for the CNF instances that were too large to be converted to d-DNNF. These generated programs still follow the same logic — simply unoptimized SAT-solving enumeration techniques. As these instances have many more solutions, this enumeration is not feasible in practice. Thus, the generated programs ran out of memory within seconds.

## 4.4 ANALYZING THE THINKING TOKENS

We also analyzed the number of reasoning tokens produced by our different experiments. Figure 2 shows the distribution (KDE) of the total number of reasoning tokens for five of our previous experiments using DeepSeek R1: end-to-end counting with the original instances (in CNF); end-to-end counting with the permuted instances (in CNF); end-to-end counting with the d-DNNF instances; code generation with the permuted instances (in CNF); and code generation with the d-DNNF instances. We used the Base prompts for all cases.

The distributions show interesting results. First, we notice that end-to-end counting with the permuted tasks generates more tokens than the analogous case with original instances. This reinforces our hypothesis that the model learned the solutions to the original tasks during training, and so it had to spend fewer tokens during inference to compute the solution. Second, end-to-end counting using d-DNNF does seem to reduce the computational effort necessary. Compared to the permuted case (Table 2), it yielded fewer reasoning tokens on average while also having higher accuracy. This supports our idea that the model could compute this simple problem using chain-of-thought. Last, but not least, it is clear that the code generation method also has an edge on the end-to-end approach in number of tokens produced.

## 5 CONCLUSION

We presented the first systematic study of LLMs on the model counting problems. Using Sudoku as a representative domain, we evaluated different strategies for model counting, ranging from direct end-to-end counting to exploiting tractable fragments and code generation. Our experimental results reveal that, while LLMs occasionally succeed at estimating solution counts, their performance is fragile to even small changes in the encoding. This suggests that memorization plays a non-negligible role in their performance.

At the same time, our experiments presented two interesting results. First, LLMs appear to benefit of inputs from tractable fragments (e.g., d-DNNF) even without being told explicitly about their tractability or that the given formula was in this format. Second, code generation seems an effective strategy for such tractable cases: all tested LLMs can produce counting algorithms that, although imperfect, solve a majority of instances correctly.

Overall, our results suggest that LLMs are not yet reliable tools for combinatorial counting, but they already exhibit intriguing behaviors. In the future we plan to test model counting with LLMs on inputs that are not grounded on logic. For example, sticking to Sudoku as an example, we would like to study what happens if we give the LLM receives a picture of the puzzle, or a textual description of the problem. Additionally, we want to study how our results extend to other fragments of model counting, such as weighted model counting.

## REPRODUCIBILITY STATEMENT

We have taken several steps to guarantee the reproducibility of our results. All main definitions (model counting problem, encodings of Sudoku into CNF and d-DNNF, and experimental setups) are described in detail in Sections 2 and 4 of the submission. We also give pointers to the original results (like theorems and definitions) and to the original source of the benchmark instances. The source code used for our experiments is included in the supplementary material. The code is sufficient for reproducing the experiments on publicly available Sudoku instances from the Model Counting Competition dataset. We include one instance of each size for local reproducibility.

To avoid potential data contamination, we do not release the raw experimental logs or the exact prompts and domains used during evaluation in this submission. In case of publication, we plan to release the complete set of prompts, used instances, and additional implementation details as an appendix to ensure full reproducibility.

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
