# OpenReview forum: "Can Large Language Models Help with Model Counting?"
_ICLR.cc/2026/Conference — Submitted to ICLR 2026_

### Official Review · Reviewer_EZFs · 2025-10-28

**Soundness:** 2
**Presentation:** 2
**Contribution:** 2
**Rating:** 2
**Confidence:** 4

**Summary:**

The authors study the problem of model counting i.e, computing multiple solutions for different problems using LLMs. The authors study this for combinatorial problems like **sudoku** and show how LLMs are fragile to simple problem modification

**Strengths:**

1. Solution multiplicity and studying solution multiplicity with respect to LLMs is an unexplored area that the authors have shed light through their work

**Weaknesses:**

1. The authors have restricted their study to only a single domain of problems which is Sudoku, there are multiple other domains that can be further explored i.e, Futoshiki, N-Queens etc. There are a bunch of problems from the SAT competition benchmark. This would have made the study more thorough and rigourous
2. Including analysis of linearly increasing the number of solutions ie from 1 to n and then investigating the performance for at each level for different sizes of sudoku would have been much more interesting
3. There is a only a mention of using LLM's to generate "python code" that computes the exact model count improves the performance but there is no mention of what kind of code was the LLM asked to generate. There are no examples of the code that the LLM has generated even in the appendix and supplementary material which makes it hard to evaluate.
4. There is no error analysis in cases where the LLMs fail when the problem is permuted, what errors during the solving process does the LLM make to make solid conclusions.

**Questions:**

Please address the above weaknesses

---

### Official Review · Reviewer_i1vC · 2025-10-29

**Soundness:** 2
**Presentation:** 3
**Contribution:** 1
**Rating:** 2
**Confidence:** 4

**Summary:**

This paper investigates the capabilities of large language models (LLMs) in solving the model counting problem, which involves determining the exact number of solutions to combinatorial problems. The authors use Sudoku puzzles as a case study to evaluate LLMs' performance in counting solutions. They find that while LLMs show some initial success, their performance is inconsistent and sensitive to changes in problem encoding. The study also explores the impact of different problem representations, particularly transforming problems into d-DNNF (deterministic decomposable negation normal form) formulas, which simplifies the counting process. The results indicate that LLMs can handle simpler problems better when provided with structured representations, although their performance remains unreliable. Additionally, the paper examines LLMs' ability to generate Python code for model counting, finding that they are more effective at producing code that can accurately perform the counting task than solving it directly. Overall, the research highlights both the potential and limitations of LLMs in addressing model counting challenges.

**Strengths:**

The paper investigates an interesting problem for LLM reasoning.

**Weaknesses:**

My general tendency is to reject this paper. This paper seems to be a technical report rather than a research paper. The experiments are not thorough enough, and the conclusions drawn are not particularly surprising or novel. The paper lacks depth in its analysis and does not provide significant insights into the capabilities of LLMs in model counting. Particular, as LLMs already struggle with simple arithmetic reasoning, it is not surprising that they would struggle with model counting, which is a more complex task (the answer would be exponentially large in general). The idea of transforming problems into d-DNNF formulas is interesting, but the paper does not explore this avenue in sufficient detail to provide meaningful conclusions. Additionally, the observation that LLMs can generate code to perform model counting is expected, given their proficiency in code generation tasks and #SAT is a well-studied problem in computer science. LLMs probably have seen many example codes for #SAT during training (the authors also mentioned the code generated utilizes existing libraries). Overall, while the paper touches on an interesting topic, it falls short in terms of depth, novelty, and significance, making it difficult to recommend for acceptance in a top-tier conference.

**Questions:**

Why did Gemini 2.5 Flash perform better on CNF than d-DNNF when using Python code?

---

### Official Review · Reviewer_EADx · 2025-10-30

**Soundness:** 3
**Presentation:** 3
**Contribution:** 1
**Rating:** 2
**Confidence:** 5

**Summary:**

This paper conducts an analysis of LLMs' abilities to perform model counting in multiple different settings (direct prompting, distinct input formula encodings, and code generation). In particular, the paper explores performance variations between the different settings, identifying key performance characteristics of the underlying AI.

Overall, the paper reveals that current models are not well-suited for a task as complex as model counting. First, there is substantial memorization, as evidenced by performance drop-off after dataset permutation. Second, the models (Gemini and DeepSeek) are not able to reliably solve instances regardless of configuration. Nonetheless, there are promising signs, namely that models do improve when the input representation is simplified (d-DNNF) and when attempting to solve instances through coding approaches.

**Strengths:**

- This is a relevant and important problem setting, which deserves to be explored.
- The approach used in this paper is sound and the results align with expectations given the complexity of the problem.

**Weaknesses:**

Unfortunately, I do not see any significant contributions being made in this paper that improve our understanding of LLMs' abilities for model counting, as I think the current experimental setup is limited and not nuanced enough on both the data and modellng side so as to deliver novel insights. Indeed, the paper highlights the limitations of LLMs in this highly complex problem, but these insights are to be expected in my opinion given the mostly straightforward set of approaches. For such complex problems, the real interest lies in whether LLMs can be augmented, e.g., a FunSearch analog for counting, in a way that would address these shortcomings, or better explain when and how these models can better align with this setting. Naturally, model counting is very complex, and the expectation on my end is not for the authors to build a functional LLM model counting system. However, the current set of baselines are mostly off-the-shelf, and don't attempt more curated methods to support the LLMs, which I believe are where the main scientific curiosities lie. Moreover, I think that including other settings for evaluation (other input encodings and benchmark suites) would strengthen this work.

As suggestions to improve this work, I recommend that the paper explore introducing algorithmic elements into their LLM analysis. Off the shelf solutions were always unlikely to achieve results in such a complex setting, but it would be interesting to see LLMs, supported by, e.g., sampling heuristics from the literature, inform algorithmic decisions (e.g., LLMs as heuristics within existing algorithms) as an intermediate and finer-grained analysis level. Going further towards more LLM control, one can also explore more settings, namely those that admit probabilistic approximation (DNF), to see if LLMs can learn more sampling-oriented behaviors, or even the established fully polynomial randomized approximation schemes (FPRAS). This would both give you more benchmarks and explore whether LLMs are better adept at identifying or solving a more probabilistic line of behavior  Approaches like FunSearch combining LLMs within a metaheuristic are also interesting. All in all, the analysis in this paper needs to be much deeper and richer on both the model and benchmark side to make a compelling case.

As a final suggestion, I also strongly recommend that the authors consider other metrics for model counting than accuracy. Given that this is a quantitative setting, pure accuracy is likely not a nuanced enough metric to explore the reliability of LLMs, particularly given that a good ballpark of the model count can, e.g., be a very useful point from which to initialize symbolic solvers in some settings. I therefore suggest that the authors include average relative deltas (estimate vs exact answer as a multiplicative ratio, to normalize against instance sizes) and likelihood probabilities of falling within multiple error percentages as a more granular, nuanced mode of analysis.

**Questions:**

None, please see the above weaknesses section.

---

### Official Review · Reviewer_aCLh · 2025-10-31

**Soundness:** 2
**Presentation:** 4
**Contribution:** 2
**Rating:** 4
**Confidence:** 4

**Summary:**

The work studies the capabilities of LLMs (DeepSeek R1, Gemini 2 Pro, and Gemini 2 Flash) in model counting, using Sudoku puzzle instances. The work evaluates how good LLMs are in counting solutions for existing CNF benchmarks, prompting with a reference to Sudoku and without a reference to Sudoku. The results are OK. However, when performing the same evaluation on transformed CNF instances, the results become much poorer, indicating the problem instances may have been part of the training process. The work also tests on d-DNNF instances instead of CNF instances of the Sudoku puzzle. In such a format, a simpler evaluation is possible, and the LLMs indeed tend to perform better once again. When prompted to generate Python code to perform the model counting, results become very good (80-90% for d-DNNF). However, this was only tested on those instances that were possible to convert into d-DNNF.

**Strengths:**

The work is written clearly, very understandable.

Studying the reasoning capability of LLMs is relevant.

The study is sound, except that it is only performed on CNFs of sudoku puzzles.

**Weaknesses:**

While the study of the reasoning capability of LLMs is relevant, combinatorial problems such as model counting are expectingly best solved using efficiently designed solvers. The work studies multiple aspects (like a transformation of a CNF, or a d-DNNF) but the results are mostly unsurprising and not significant. Furthermore, the study is performed on only CNFs of a Sudoku (although the results are likely similar for other CNFs).

**Questions:**

Q1) How many Sudoku instances were used? How many instances were used in the d-DNNF experiment, and could you elaborate on the number of variables and d-DNNF size?

Q2) "As the conversion from CNF to d-DNNF might take exponential time, only half of the instances of our dataset could be converted. These correspond to the smallest and easiest half of the set." What time-out was used for this?

Q3) "the inference to the correct answers is sound". Can you say something more about this inference? Was it applying a DPLL-style of reasoning?

## Remarks

For CNF, DIMACS was used. The work does not mention the format for the d-DNNFs.

Typos:
* "the model is solved a Sudoku problem"
* "to for a"

---

### Author Response · Authors · 2025-12-03

Re Reviewer aCLh:

Q1) We used the entire dataset of available Sudoku instances that have been submitted to the model counting competition, see the cited Zenodo dataset.

Q2) We only used those instances that could be d-DNNF decomposed within 25 minutes. Note, however, that time was not an issue. Mostly the sizes of some computed d-DNNFs were too large for the LLMs to process.

Q3) No, DPLL is by far not enough, and in particular not enough for counting. The LLMs were arguing via the circuit, as in d-DNNF counting. We were surprised how well these LLMs were doing.

Re Reviewer EADx:
> Indeed, the paper highlights the limitations of LLMs in this highly complex problem, but these insights are to be expected in my opinion given the mostly straightforward set of approaches.

Our experimental results are more nuanced than this. We study that different inputs produce different levels of performance, and some of them are quite surprising. For example, the fact that LLMs can solve d-DNNF problems reliably (between 80-90% of solved tasks) is remarkable. There are polynomial-time algorithms for these formulas, but this is far from widespread knowledge. We believe that our paper provides a baseline for research on model counting using LLMs.

> Going further towards more LLM control, one can also explore more settings, namely those that admit probabilistic approximation (DNF), to see if LLMs can learn more sampling-oriented behaviors, or even the established fully polynomial randomized approximation schemes (FPRAS). This would both give you more benchmarks and explore whether LLMs are better adept at identifying or solving a more probabilistic line of behavior Approaches like FunSearch combining LLMs within a metaheuristic are also interesting.

This suggestion is already included in the submitted version. We study the performance of LLMs in the d-DNNF fragment, a well-studied and practically relevant fragment of SAT where model counting is tractable. This is a technique commonly used in model-counting competitions (i.e., try to translate to d-DNNF and used specialized algorithms). We believe that d-DNNF is much more practically relevant and empirically studied (e.g., many state-of-the-art model counters exploit this fragment) than your suggested analyses.

Re Reviewer i1vC:

Note that counting combinatorial solutions and arithmetic are quite different and there are also approximation algorithms (FPRAS) for counting on fragments. We study that different inputs produce different levels of performance, and some of them are quite surprising on the positive note. For example, the fact that LLMs can solve d-DNNF problems reliably (between 80-90% of solved tasks) is remarkable. There are polynomial-time algorithms for these formulas, but this is far from widespread knowledge. We believe that our paper opens an interesting direction for research on model counting using LLMs without just connecting agents.

Re Reviewer EZFs:

Answering to your questions in order:

Q1) Sudoku is a very interesting problem in this setting. First, the space of problems is more fine-grained and detailed. Second, Sudoku has been actively studied and was used in state-of-the-art benchmark sets to compare solvers. Last, but not least, Sudoku has a community interested in Sudoku by itself.

Q2) Linearly increasing the number of solutions does not correlate with making the problem harder. Recall that counting is not enumeration without outputting. This can be seen in practice, even with Sudoku: the problem with most solutions is the Sudoku with zero starting clues. But this problem is much easier to solve for an LLM than a problem with a few well-placed clues, as the empty grid has a known number of solutions.

Q3) The Python code generated receives the problem file as input. It must parse and solve the problem by outputting the answer. We will include a few examples in our next iteration.

Q4) We will discuss the errors committed by the LLMs in the next iteration.

---

### Meta-Review · Area_Chair_Gjh8 · 2026-01-03

**Summary:**

The paper studies use of LLMs for model counting. The paper received a 2,2,2,4. All reviewers strongly or weakly suggest rejecting the paper in this iteration. Main concern appears to be the fact that none of this is particularly surprising – that the LLMs do not do as well on model counting. The one interesting tidbit is that d-DNNF representation is likely better for LLMs than CNF. But, since d-DNNF have been constructed only for 50% of the CNF formulas, it is not clear whether the timeout in the conversion process itself represents an easier subset of model counting queries. If this correlation were to exist then this result was would be re-interpreted as LLMs can help with the easier subset of problems… not that LLMs can help with d-DNNF problems. The authors did not respond to this question.

Overall, for publication at a top venue such as ICLR the depth of the contribution, novelty and significance of work should be much higher than what’s currently in this paper. The reviewers suggest many different ways in which the work may be deepened.

**Reviewer Concerns:**

Main concern: nothing surprising in this paper.
Many other concerns such as only 1 domain studied, correlation between hardness of problem and d-DNNF conversion not explored, no error analysis, etc.

**Reviewer Scores:**

I don’t expect reviewers to have changed the scores much given that the key issue with the paper would have remained unresolved. Maybe 2 would have become 3 or something but that does not summarily change the overall decision on the paper.

---

### Decision · Program_Chairs · 2026-01-26

Reject